# HYPERKAN: A PLUG-AND-PLAY TOOL FOR PERSONALIZED WEIGHTS GENERATION

## ABSTRACT

Personalized weight generation is vital for adapting models to tailored patterns in real-world applications. However, existing methods struggle to capture nuanced feature variations across heterogeneous data distributions, resulting in suboptimal personalization performance. In this paper, we introduce HyperKAN, a plug-and-play personalized weights generator that integrates Kolmogorov-Arnold Networks (KANs) for capturing feature variations among clients and enhancing personalization capacity. We design a novel Personalized Federated Learning (pFL) framework that embed HyperKAN, enabling tailored model aggregation for each client with faster convergence. Our evaluations on four datasets present HyperKAN's versatility and effectiveness, achieving up to 48% higher accuracy than state-of-the-art methods. In a nutshell, HyperKAN offers a highly adaptable solution for enhancing model personalization, particularly in non-IID scenarios that challenge traditional weight generation approaches.

## 1 INTRODUCTION

The increasing demand for personalized models necessitates tailored machine learning models in various scenarios (*e.g.*, recommendation system Yang et al. (2024); Zhang et al. (2024), unmanned aerial vehicle cluster Soltani et al. (2023); Qu et al. (2022), and Artificial Intelligence of Things (AIoT) Kamani et al. (2023); Imteaj et al. (2021); Zhang et al. (2022)). These scenarios commonly face heterogeneous data distributions that create obstacles in producing optimal personalized weights. This underscores the significance of developing method that can efficiently handle heterogeneous data during personalized weight generation.

In current research, personalized weight generation is divided into two categories: implicit and explicit. Implicit methods optimize initial models to approximate personalized models (Li et al. (2020b), Arivazhagan et al. (2019), Li et al. (2021b)). Building on this foundation, Ye et al. (2023) introduces a method that constructs collaboration graphs to coordinate the optimization process of personalized models. Despite this, Implicit personalized weight generation often requires an extensive number of model updates and precise optimization target design. In contrast, explicit methods directly generate personalized weights or model parameters for personalized objectives through additional designed generators or dynamic mechanisms, such as Zhang et al. (2024) and Shamsian et al. (2021).

Recently, Hypernetwork Ha et al. (2017) have emerged as a promising explicit solution for personalization by generating specific model updates Du et al. (2024); Brock et al. (2018); Zhang et al. (2019). Hypernetwork-based pFl leverage parameter variations between training iterations to generate personalized model weights, capturing the underlying relationships and similarities across different model training processes Du et al. (2024). Furthermore, Hypernetwork often are utilized into the Personalized Federated Learning (pFL) Ma et al. (2022); Shamsian et al. (2021). However, existing Hypernetwork-based pFL methods, such as pFedLA Ma et al. (2022) and pFedHN Shamsian et al. (2021), face the challenge of adequately capturing the intricate non-linearities present in heterogeneous data distributions. These methods rely on MLPs to extract features during training, which struggle to capture complex relations of training parameters with heterogeneous data Smith et al. (2017); Li et al. (2020a), limiting their model expressive capability in highly heterogeneous settings. Specifically, the representation learning capacity of MLPs is inherently constrained by their use of fixed activation functions on linear transformations. In contrast, Kolmogorov-Arnold

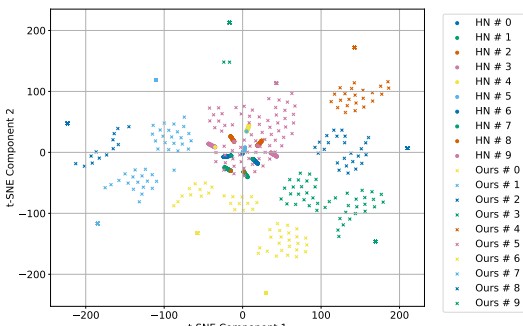

Figure 1: The parameter variation features extracted through HyperKAN and the Hypernetwork. Each point represents the parameter variation of one communication round.

Networks (KANs) have recently been shown to possess superior representation capabilities by replacing linear weights with learnable, spline-based activation functions, enabling them to approximate complex functions with greater precision and parameter efficiency Toscano et al. (2025); Wang et al. (2025). This distinction is crucial for personalization. Figure 1 demonstrates this point in that the features extracted by MLP-based Hypernetworks are clustered together, making the variations in personalized parameters across clients unable to be distinguished effectively. Such excessively similar intra-client similarity can lead to the failure of generating personalized weights.

To address the challenges of highly heterogeneous data and underdeveloped model personalization capabilities, we introduce HyperKAN, a plug-and-play personalized weights generator that integrates Kolmogorov-Arnold Networks (KANs) Liu et al. (2025), designed to enhance the server's expressivity to capture the personalization of client models. HyperKAN operates on the server-side, generating personalized aggregation weights that adapt to heterogeneous data distributions. We demonstrate HyperKAN's versatility and effectiveness within our pFL framework called KpFL. In summary, the main contributions are summarized as follows:

- We propose HyperKAN, a novel personalization module that leverages KANs to generate personalized aggregation weights in pFL. HyperKAN captures fine-grained variations in client model parameters, enabling more expressive client weight generation.
- We introduce KpFL, a new pFL framework that integrates HyperKAN with a critical block aggregation mechanism, which selectively aggregates key network components to accelerate convergence and improve training efficiency.
- We demonstrate its effectiveness across different data patterns, achieving up to 48% higher accuracy than state-of-the-art methods on four real-world datasets, highlighting HyperKAN's potential for broad applicability.
- We provide an empirical analysis of HyperKAN's powerful personalization capabilities, demonstrating that HyperKAN produces more dispersed and distinctive parameter variations than MLPs. This dispersion facilitates better client differentiation and supports more effective personalized aggregation.

## 2 PRELIMINARY

**KAN** Liu et al. (2025), employed in HyperKAN, offer a novel method to represent complex functions. The underlying Kolmogorov-Arnold theorem states that any continuous multivariate function $f : \mathbb{R}^d \to \mathbb{R}$ can be decomposed into a finite sum of continuous univariate functions:

$$f(x_1, x_2, \ldots, x_d) = \sum_{q=1}^{2d+1} \phi_q \left( \sum_{p=1}^{d} \psi_{pq}(x_p) \right), \tag{1}$$

where $\phi_q$ and $\psi_{pq}$ are continuous univariate functions. This decomposition enables KANs to capture intricate relationships by combining simpler, univariate transformations. A KAN layer implements

a generalized form of representation:

$$h^{(l+1)} = \sigma \left( \sum_{q=1}^{2d+1} \phi_q^{(l)} \left( \sum_{p=1}^{d} \psi_{pq}^{(l)}(h_p^{(l)}) \right) \right), \tag{2}$$

where $h^{(l)}$ represents the activations at layer $l$, $\phi_q^{(l)}$ and $\psi_{pq}^{(l)}$ are learnable univariate functions (parameterized as neural networks in our implementation), and $\sigma$ is a non-linear activation function (SiLU in HyperKAN). HyperKAN utilizes multiple stacked KAN layers, enabling hierarchical feature extraction and capturing complex relationships between client models. The forward pass through a multi-layer KAN with $L$ layers is:

$$
\begin{aligned}
h^{(L)} = \sigma &\left( \sum_{q=1}^{2d+1} \phi_q^{(L-1)} \left( \sum_{p=1}^{d} \psi_{pq}^{(L-1)}(h_p^{(L-1)}) \right) \right) \\
&\cdots \\
\sigma &\left( \sum_{q=1}^{2d+1} \phi_q^{(1)} \left( \sum_{p=1}^{d} \psi_{pq}^{(1)}(x_p) \right) \right).
\end{aligned}
\tag{3}
$$

## 3 DESIGN OF HYPERKAN

HyperKAN maps the parameter variations of each client to a unified embedding matrix, then extracts the features of these variations and generates the personalized weight.

HyperKAN's architecture is based on a hierarchical composition of $L_H$ KAN layers, enabling it to capture the subtle variations in the complex model parameters. Formally, HyperKAN, denoted as $h_{KAN}$, is defined as:

$$h_{KAN}(\Delta_i) = (\Phi^{(L_H-1)} \circ \cdots \circ \Phi^{(0)})(\Delta_i), \tag{4}$$

where each $\Phi^{(l)}$ denotes a KAN layer, and $\circ$ denotes composition. Each KAN layer within Hyper-KAN further comprises $2d + 1$ parallel branches, as defined in Equation 2 (where $d$ is the input dimension), enhancing its capacity to discern subtle relationships between client model parameter variations. We use the vanilla parameters and KAN settings for each KAN layer. Within each KAN layer, the $\phi_q$ and $\psi_{pq}$ functions are implemented as two-layer fully connected neural networks with neurons that vary with depth and nonlinear activation functions per layer. This configuration, de-termined empirically, offers a suitable trade-off between expressivity and computational cost. The multi-layer structure of the $\phi_q$ and $\psi_{pq}$ networks further enhances HyperKAN's ability to capture intricate, hierarchical relationships between client models, leading to more precise and adaptable aggregation. Each $\psi_{pq}^{(l)}$ acts as a feature transformer on the input data, while $\phi_q^{(l)}$ aggregates these transformed features, mimicking the additive nature of the Kolmogorov-Arnold representation, to compute the output $h^{(l+1)}$. The forward pass with $L$ layers can be represented by recursively ap-plying Equation 2. HyperKAN extracts features based on block-wise variation $(\Delta_i^{ln})$ in model parameters, enabling accelerated convergence and personalization by aggregating clients with simi-lar parameter update features at the block granularity. HyperKAN parameterizes the client variations through $v_i$ and performs feature extraction. Since clients with similar distributions exhibit similar variations during training, the features extracted by HyperKAN can be used to generate weights in the feature space. As shown in Figure 3, the trainable embedding vectors $v_i$ are used as inputs to a multilayer perceptron composed of KAN layers, which outputs features of parameter variation. By learning the model parameter dynamic features from different clients, the HyperKAN ultimately generates the corresponding weights.

For block-wise aggregation weights $\alpha_i^{ln}$, the calculation method is given in Equation 5.

$$\alpha_i^{ln} = \frac{\sigma \left( \text{KAN}_i(\epsilon_i^{ln}) \right)}{\sum_{i=1}^{N} \sigma \left( \text{KAN}_i(\epsilon_i^{ln}) \right)}, \tag{5}$$

where $\epsilon_i^{ln}$ is the feature of parameter variations extracted by Equation 4.

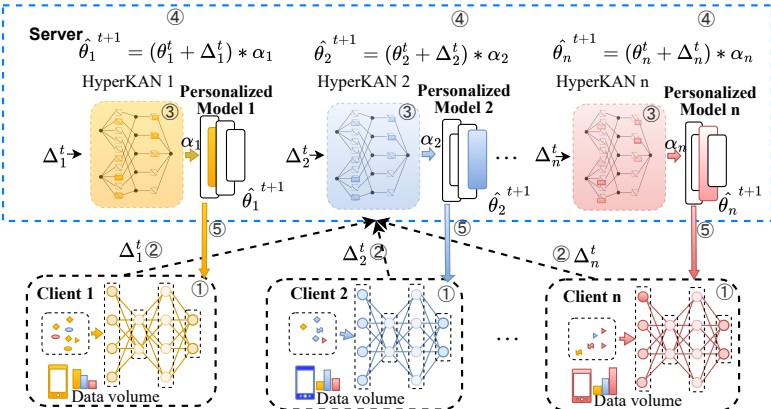

Figure 2: The architecture of KpFL. ①Local train. ②Upload parameter variations. ③Generate personalized aggregation weights through HyperKAN. ④Perform personalized weighted aggregation. ⑤Distribute personalized model.

## 4 KpFL: HyperKAN for Personalized Federated Learning

### 4.1 Problem Formulation

Personalized Federated Learning aims to train customized models for individual clients in a decentralized manner, respecting data privacy. A key challenge is statistical heterogeneity, particularly prevalent in the real world with diverse user populations and data distributions Ju et al. (2024b;a). Formally, in pFL, we aim to learn personalized models, $\{\theta_i\}_{i=1}^{N}$, for $N$ clients, each with a local dataset $\mathcal{D}_i$. The objective is to minimize the total loss across all clients:

$$\min_{\{\theta_i\}_{i=1}^{N}} \sum_{i=1}^{N} \mathcal{L}(\theta_i; \mathcal{D}_i), \tag{6}$$

where $\mathcal{L}(\theta_i; \mathcal{D}_i)$ is the client $i$'s model loss on its local data.

### 4.2 KpFL Framework

We seek a solution that effectively addresses statistical heterogeneity while minimizing client-side overhead, enabling practical deployment in diverse, potentially resource-constrained, settings. HyperKAN is designed to enhance existing pFL methods by generating personalized aggregation weights on the server. Figure 2 is the KpFL framework which illustrates how HyperKAN integrates into a typical pFL pipeline. Algorithm 1 illustrates the complete training process of the KpFL.

During initialization, KpFL uploads the model's full parameters, and subsequently, only the variations of the local model parameters ($\Delta_i^{t+1} = \theta_i^{(t+1)} - \theta_i^{(t)}$) are uploaded. The server obtains the latest client model using Equation 7.

$$\theta_i^t = \theta_i^{t-1} + \Delta_i^t. \tag{7}$$

### 4.3 The Training Process of KpFL

The objective function of KpFL can be derived from Eq. 6 to

$$\arg\min_{V,\Phi} \sum_{i=1}^{N} \mathcal{L}_i(\theta_i * HN_i(v_i; \phi_i)), \tag{8}$$

the optimization problem transfers to the HyperKAN's embedding $v_i$ and parameters $\phi_i$. We can have the gradient of $v_i$ and $\phi_i$ from Eq. 8:

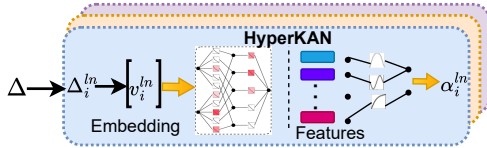

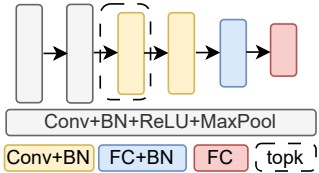

Conv+BN+ReLU+MaxPool

Conv+BN | FC+BN | FC | topk

Figure 3: HyperKAN architecture.

Figure 4: Block-wise method.

$$\nabla_{v_i}\mathcal{L}_i = (\nabla_{v_i}\hat{\theta}_i)^T\nabla_{\hat{\theta}_i}\mathcal{L}_i$$
$$= [\,\theta_i * \nabla_{v_i}h_{KAN_i}(v_i;\phi_i)]^T\nabla_{\hat{\theta}_i}\mathcal{L}_i, \tag{9}$$

$$\nabla_{\phi_i}\mathcal{L}_i = (\nabla_{\phi_i}\hat{\theta}_i)^T\nabla_{\hat{\theta}_i}\mathcal{L}_i$$
$$= [\,\theta_i * \nabla_{\phi_i}h_{KAN_i}(v_i;\phi_i)]^T\nabla_{\hat{\theta}_i}\mathcal{L}_i, \tag{10}$$

where $\nabla$ is the gradient, $\nabla_{\hat{\theta}_i}\mathcal{L}_i$ can be obtained from client $i$.Therefore, the gradient update method is as follows:

$$\Delta v_i = (\nabla_{v_i}\hat{\theta}_i)^T\Delta\theta_i$$
$$= [\,\theta_i * \nabla_{v_i}h_{KAN_i}(v_i;\phi_i)]^T\Delta\theta_i, \tag{11}$$

$$\Delta\phi_i = (\nabla_{\phi_i}\hat{\theta}_i)^T\Delta\theta_i$$
$$= [\,\theta_i * \nabla_{\phi_i}h_{KAN_i}(v_i;\phi_i)]^T\Delta\theta_i. \tag{12}$$

where $\Delta\theta_i$ is the variation of model parameters in client $i$ after local training.

### 4.3.1 CRITICAL BLOCK AGGREGATION MECHANISM

In pFL, it is generally undesirable to learn the parameter variations of the entire client model network Shamsian et al. (2021). It is preferable to output each part of the client model network. To accelerate aggregation while minimizing the impact of the aggregated model on local parameters, we introduce a critical block aggregation mechanism. As shown in Figure 4, for the parameters on a client, the top-$k$ chunks are selected as the critical blocks. Hierarchical aggregation methods have been proven to be beneficial for generating personalized models in pFL Ma et al. (2022); Shamsian et al. (2021); Chen et al. (2022).

In KpFL, the model parameters of client $i$ is obtained by weighted aggregation according to aggregation matrix, $\alpha_i$:

$$\alpha_i = \begin{bmatrix} \alpha_i^{l1,1} & \alpha_i^{l2,1} & \cdots & \alpha_i^{ln,1} \\ \alpha_i^{l1,2} & \alpha_i^{l2,2} & \cdots & \alpha_i^{ln,2} \\ \vdots & \vdots & \ddots & \vdots \\ \alpha_i^{l1,N} & \alpha_i^{l2,N} & \cdots & \alpha_i^{ln,N} \end{bmatrix}. \tag{13}$$

Let $\alpha_i^{ln}$ denote the aggregation weight vector for the $n$-th block of client $i$, and let $\alpha_i^{ln,N}$ represent the aggregation weight corresponding to client $N$ in the $n$-th block. Across all $n$ blocks, the condition $\sum_{j=1}^{N}\alpha_i^{ln,j} = 1$ must hold. The personalized parameters $\theta_i$ distributed to the client are calculated using Equation 14.

$$\hat{\theta}_i = \{\hat{\theta}_i^{l1}, \hat{\theta}_i^{l2}, \dots, \hat{\theta}_i^{ln}\} = \{\theta^{l1}, \theta^{l2}, \dots, \theta^{ln}\} * \alpha_i, \tag{14}$$

where $\hat{\theta}_i^{ln}$ can also be expressed as:

$$\hat{\theta}_i^{ln} = \sum_{j=1}^{N}\theta_j^{ln}\alpha_i^{ln,j}. \tag{15}$$

Before executing the parameter distribute, we need to update the parameters of the critical blocks into the parameters to be dispatched. Critical blocks refer to the blocks with large $\Delta$ values, as these parameters have undergone more significant variations during training. For the parameters of the critical blocks, we aggregate them, while the parameters of other blocks remain unchanged.

$$\hat{\theta}_i = \hat{\theta}_i \odot \{\alpha_i^{l1}, \dots, \alpha_i^{lk}, \alpha_i^{ln}\}, \tag{16}$$

where $\hat{\theta}_i$ is the personalized parameters, and $\odot$ represents the Hadamard product.

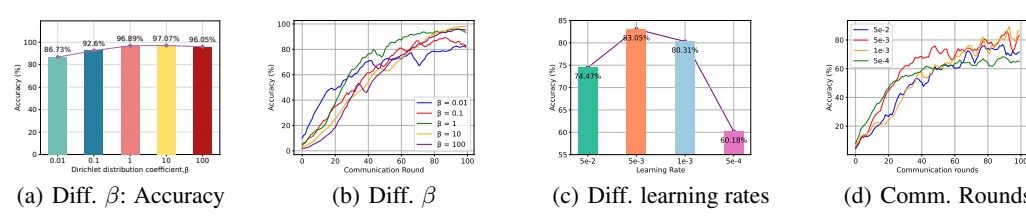

(a) Diff. $\beta$: Accuracy     (b) Diff. $\beta$     (c) Diff. learning rates     (d) Comm. Rounds

Figure 5: Sensitivity Analysis of KpFL on CIFAR-100

### 4.3.2 CLIENT MODEL UPDATE

Following the block-wise aggregation process, each client $i$ receives its updated model parameters, $\hat{\theta}_i$, from the server. These aggregated parameters serve as the initial point for the next round of local training on the client's device. The local update is formalized as:

$$\theta_i \leftarrow \hat{\theta}_i - \eta \nabla_{\hat{\theta}_i} \mathcal{L}(\hat{\theta}_i; \mathcal{D}_i), \tag{17}$$

where $\eta$ is the learning rate, and $\nabla_{\hat{\theta}_i} \mathcal{L}$ denotes the gradient of the loss function with respect to the model parameters. This iterative process ensures each client's model consistently benefits from the collective knowledge shared across the federation, while remaining tailored to its specific data distribution.

Table 1: Accuracy on four datasets partitioned using a Dirichlet distribution with $\beta = 0.1$ .

| Dataset | FMNIST | | EMNIST | | CIFAR-10 | | CIFAR-100 | |
|---|---|---|---|---|---|---|---|---|
| # Clients | 10 | 20 | 10 | 20 | 10 | 20 | 10 | 20 |
| FedAvg | 86.9 | 93.9 | 92.3 | 92.8 | 75.4 | 77.8 | 47.5 | 47.8 |
| MetaFed | 96.7 | 95.1 | 94.8 | 94.3 | 90.1 | 86.1 | 44.2 | 52.1 |
| Elastic | 81.5 | 82.4 | 85.1 | 84.4 | 63.1 | 63.9 | 29.6 | 30.5 |
| Floco | 94.2 | 93.4 | 94.1 | 93.4 | 88.4 | 86.5 | 62.3 | 55.9 |
| pFedLA | 95.8 | 92.7 | 88.6 | 88.7 | 90.4 | 87.2 | 61.3 | 53.5 |
| KpFL(Ours) | 98.8↑2.1% | 96.1↑2.3% | 96.8↑2.1% | 98.9↑4.8% | 99.1↑9.6% | 94.1↑7.9% | 92.6↑48% | 82.6↑47% |

## 5 EVALUATION

### 5.1 EXPERIMENTAL SETUP

**Datasets.** We evaluate HyperKAN on four benchmarks: Fashion-MNIST (FMNIST) Xiao et al. (2017), CIFAR-10 Krizhevsky (2009), CIFAR-100 Krizhevsky (2009), and EMNIST Cohen et al. (2017). Further implementation details are provided in Appendix A.

**Baselines.** We benchmark HyperKAN against several established pFL methods. These include the foundational FedAvg McMahan et al. (2017); methods addressing system or data challenges like Elastic Chen et al. (2023a) and Floco Grinwald et al. (2024); and advanced personalization approaches like MetaFed Chen et al. (2023b). Crucially, we also compare against pFedLA Ma et al. (2022), another method utilizing a Hypernetwork. For a fair comparison, all methods are evaluated under consistent experimental settings, with hyperparameters individually tuned for optimal performance.

**Evaluation Metrics.** We evaluate performance using accuracy, loss, and time, across varying communication rounds. Scalability is assessed by analyzing performance with increasing client numbers. The number of critical blocks $K$ in Table 1 is 2. The goal of personalized learning is to train models customized to the unique data distributions and specific needs of individual clients, ultimately leading to improved accuracy on local tasks. To evaluate this, we simulate challenging non-IID scenarios using the Dirichlet distribution, a standard and widely accepted methodology for assessing personalization performance in pFL Fallah et al. (2020); Li et al. (2021a).

## 5.2 PERFORMANCE EVALUATION

### 5.2.1 ACCURACY COMPARISON ON DIFFERENT SETTINGS

Table 1 presents the accuracy achieved by HyperKAN and the baselines on the four benchmark image datasets. As the table clearly shows, integrating HyperKAN consistently improves performance across all datasets and client configurations. On the CIFAR-10 dataset, KpFL achieves the highest accuracy of 99.1% with 10 clients and 94.1% with 20 clients, surpassing pFedLA(Hypernetwork based) by a substantial 9.6% and 7.9%, respectively. This demonstrates HyperKAN's effectiveness in enabling personalized models, even with non-IID data. Notably, on the more challenging CIFAR-100 dataset, the integration of HyperKAN leads to significant improvements, achieving 92.6% accuracy with 10 clients and 82.6% with 20 clients, 47% higher than Floco. This 47% improvement underscores HyperKAN's ability to capture and adapt to the intricate data heterogeneity present in CIFAR-100.

Figures 11 and 12, presented in the Appendix, illustrate the accuracy curves for each method, demonstrating the consistent superiority of integrating HyperKAN in terms of both final accuracy and convergence speed. These results demonstrate the effectiveness of HyperKAN for personalized federated learning. By leveraging HyperKAN's expressiveness and strategic server-side placement, the framework can effectively handle the diverse and distributed data characteristics inherent in federated learning environments, leading to improved performance and faster convergence.

**Convergence Analysis.** Figure 13 displays the training loss curves, demonstrating HyperKAN's superior efficiency. Our method consistently achieves a faster loss reduction and a lower final loss value compared to all baselines. This rapid convergence, particularly in the initial rounds, highlights HyperKAN's ability to quickly adapt to diverse data distributions. For example, on CIFAR-100, HyperKAN's final loss of approximately 0.18 is significantly lower than the 1.2 achieved by pFedLA. This performance indicates that HyperKAN effectively captures client variations, leading to more efficient personalized aggregation.

### 5.2.2 COMPUTATIONAL EFFICIENCY AND COMPLEXITIES ANALYSIS

Figure 6 compares the time required by Hypernetwork and HyperKAN on a single local update of the client. Under the same architectural configuration, HyperKAN incurs a slightly longer update time compared to Hypernetwork. However, this time consumption is negligible compared to the local update time on the client-side. Importantly, HyperKAN's computation occurs solely on the server, minimizing client-side burden. The significant accuracy improvements achieved by incorporating HyperKAN (Table 1, Figures 11 and 12) justify this server-side computational cost. The improved accuracy, especially in non-IID scenarios, outweighs the time increment of HyperKAN.

We assume a HyperKAN with depth $L$, width $N$, grid size $G$, and spline order $k$. The model has $O(N^2GL)$ parameters. Assuming a training batch size is $B$, memory usage is $O(2^kBN^2GL)$. The operations number is $O(2^kBN^2GL)$ both for forward and backward steps. The $2^k$ factor is due to the recursive computation of order $k$ splines. Therefore, a larger spline order leads to a rapid increase in computational complexity.

Table 2 presents the average parameter amount and computational FLOPs of models between HyperKAN and an MLP-based Hypernetwork (specifically, the architecture comparable to pFedLA Ma et al. (2022)) in identical settings. While HyperKAN has more trainable parameters, its spline-based computation results in significantly lower Multiply-Accumulate Counts(MAC). This efficiency keeps HyperKAN's computational cost reasonable, as shown in Figure 6. As Table 1 shows, there is a substantial personalized performance improvement for training by a favorable trade-off.

Table 2: Computational cost of Hypernetwork vs. HyperKAN

| Plugin | FLOPs (MACs) | Parameters |
|---|---|---|
| Hypernetwork | 38.55K | 39,346 |
| HyperKAN | **0.65K** | 382,440 |

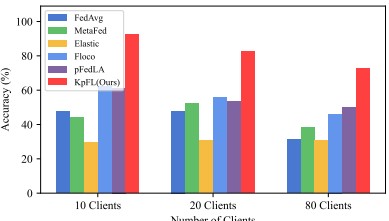
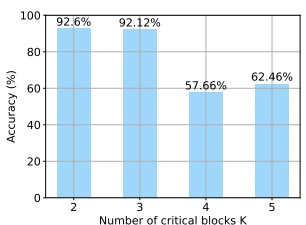

Figure 6: Time comparison

Figure 7: Scalability.

Figure 8: Different K

### 5.2.3 SCALABILITY ANALYSIS

Integrating HyperKAN consistently achieves the highest accuracy across varying client counts, demonstrating robust performance. On CIFAR-100 with 80 clients, a highly non-IID scenario, HyperKAN significantly outperforms baselines, confirming its scalability. As shown in Figure 7, KpFL maintains superior accuracy across different client numbers, highlighting its effectiveness in large-scale, non-IID settings.

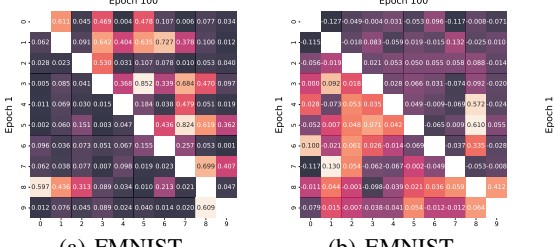
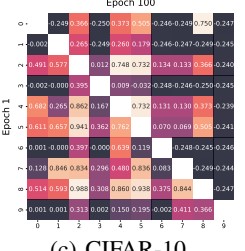
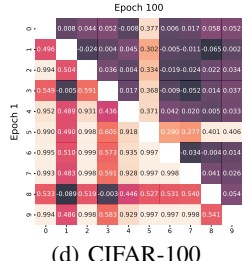

(a) FMNIST      (b) EMNIST      (c) CIFAR-10      (d) CIFAR-100

Figure 9: Model Weight Similarity Among 10 Clients. Cosine similarity heatmaps at the start and end of training illustrate how HyperKAN adaptively guides model convergence and knowledge sharing across four datasets with diverse data distributions.

### 5.2.4 ANALYSIS OF THE VARYING NUMBER OF $K$

Figure 8 illustrates the impact of varying K (the number of critical blocks aggregated) on model accuracy under non-IID CIFAR-100 with 10 clients. Optimal performance occurs at $K = 2$: excessive blocks amplify personalized parameter divergence, introducing client-specific features that exacerbate parameter drift. Notably, when $K = 2$, the first convolution block and classifier layer emerge as the most dynamically adjusted blocks during training. The first convolution block adapts to low-level feature distribution shifts across clients, while the classifier layer adjusts decision boundaries to reconcile client-specific class distributions, making them focal points for dynamic personalization in pFL.

### 5.3 SENSITIVITY ANALYSIS

We conducted sensitivity analyses to evaluate HyperKAN's robustness to varying data distributions and hyperparameter settings. Figure 5 summarizes the results on CIFAR-100. Figures 5(a) and 5(b) show HyperKAN's performance under varying degrees of non-IID data, controlled by the Dirichlet distribution coefficient $\beta$. HyperKAN maintains high accuracy across all $\beta$ values, demonstrating robustness to different levels of data heterogeneity. This adaptability is crucial for real-world deployments where data distributions can vary significantly across clients. Figures 5(c) and 5(d) analyze the influence of HyperKAN's learning rate. A learning rate of 5e-3 achieves the highest final accuracy, while 1e-3 exhibits faster initial convergence. These results stress hyperparameter tuning and HyperKAN's robustness across settings.

### 5.4 INTERPRETABILITY ANALYSIS

To understand how HyperKAN adapts to individual client data characteristics, we visualize the evolution of client model weights during training. Figure 9 presents heatmaps showing the co-

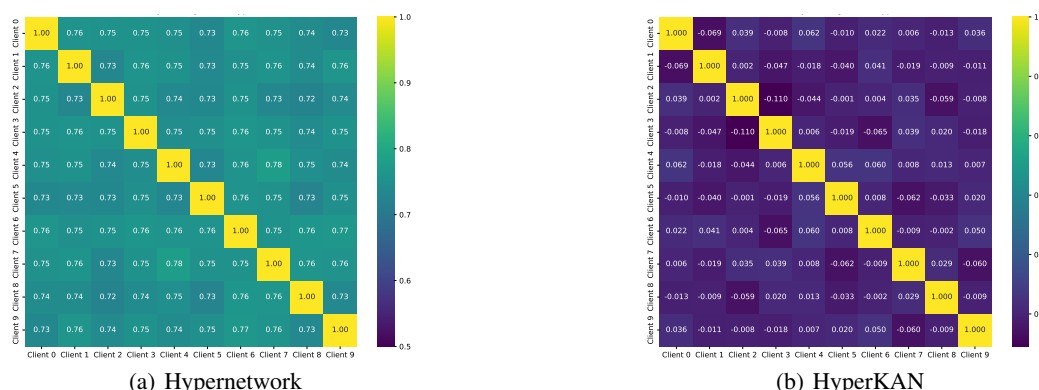

(a) Hypernetwork            (b) HyperKAN

Figure 10: The similarity of parameters in the Linear layers of each client's Hypernetwork and HyperKAN with 10 clients.

Table 3: Standard Deviation of aggregation weight $\alpha$

| # of Rounds | Hypernetworks | HyperKAN |
|---|---|---|
| 10 | 0.01 | 0.21 |
| 50 | 0.009 | 0.19 |
| 100 | 0.011 | 0.23 |

sine similarity between client models at the beginning (epoch 1) and end (epoch 100) of training in a 10-client setting. Initially, similarity varies significantly, reflecting the inherent heterogeneity of local data distributions. As training progresses, distinct clusters of clients with similar model weights emerge for FMNIST, EMNIST, and CIFAR-10, suggesting that HyperKAN facilitates effective knowledge sharing among clients with similar data. CIFAR-100, a more complex dataset, exhibits less clustering due to more specialized client models. This highlights HyperKAN's ability to adapt to diverse datasets and client distributions.

HyperKAN demonstrates enhanced personalization by effectively learning client-specific representations from heterogeneous data. As visualized in Figure 10, the high parameter similarity in a traditional hypernetwork indicates limited personalization, whereas HyperKAN maintains low similarity, reflecting diverse client models. This diversity is quantified in Table 3, where HyperKAN's generated weights show a considerably larger standard deviation. Therefore, HyperKAN's ability to amplify unique client features marks a significant improvement in adaptability over the uniform outputs of conventional hypernetworks.

## 6 CONCLUSIONS

In this paper, we present HyperKAN, a plug-and-play personalized weight generation tool that integrates Kolmogorov-Arnold Networks to capture complex dependencies via model parameter variations, thereby enhancing the capacity for personalized representation. Leveraging the expressive power of KANs, HyperKAN dynamically adapts to diverse client-specific requirements, making it particularly effective for real-world applications characterized by high variability. Integrated into our novel pFL framework, HyperKAN facilitates personalized aggregation, accelerates convergence, and effectively addresses key challenges posed by non-IID data distributions. Experimental results demonstrate the scalability and adaptability of HyperKAN, underscoring its potential to improve model personalization in resource-constrained and heterogeneous data environments. In a nutshell, HyperKAN offers a highly adaptable solution for personalized weights generation.

## REPRODUCIBILITY STATEMENT

The source code used to generate the results and figures presented in this paper is available as part of the Supplementary Material.

## ETHICS STATEMENT

This research adheres to the ICLR Code of Ethics. We are committed to fostering scientific integrity and collaboration, respecting privacy and fairness, and striving to ensure our work positively impacts society while mitigating potential harms.

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

# A    APPENDIX

## USE OF LLM

We used a Large Language Model (LLM) for grammar proofreading and language polishing in this paper.

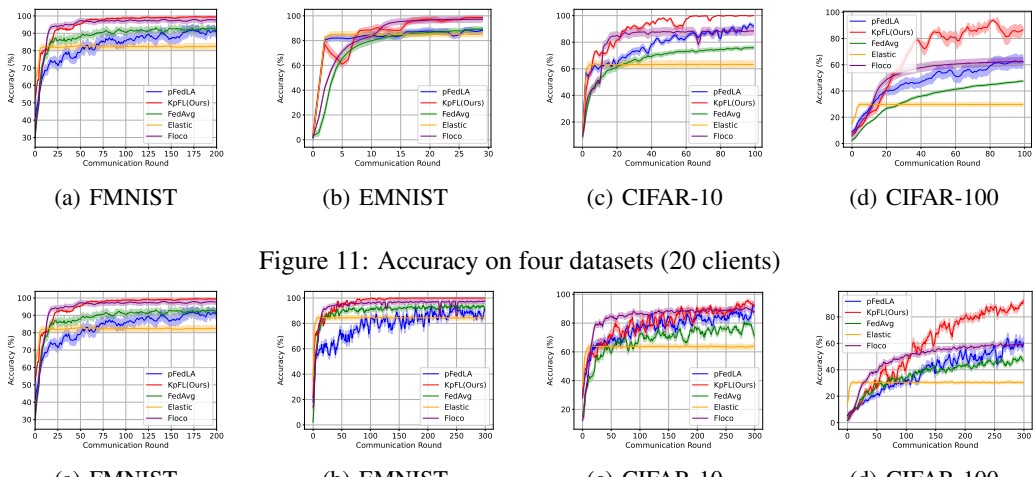

|   (a) FMNIST   |   (b) EMNIST   |   (c) CIFAR-10   |   (d) CIFAR-100   |

Figure 11: Accuracy on four datasets (20 clients)

|   (a) FMNIST   |   (b) EMNIST   |   (c) CIFAR-10   |   (d) CIFAR-100   |

Figure 12: Accuracy on four datasets (20 clients)

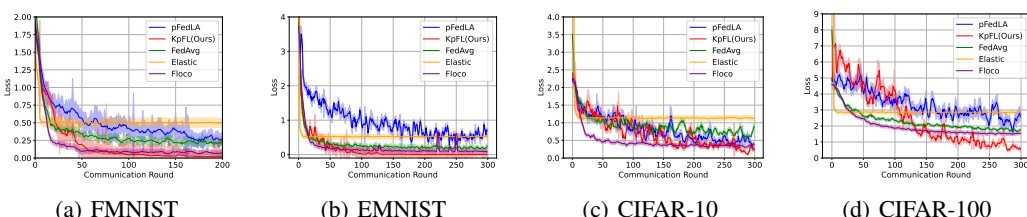

|   (a) FMNIST   |   (b) EMNIST   |   (c) CIFAR-10   |   (d) CIFAR-100   |

Figure 13: Loss on four datasets (20 clients)

**Implementation.**    All methods are implemented in PyTorch 2.3.1 and executed on a cluster (NVIDIA RTX 4090, 256 GB RAM). We use a consistent learning rate (0.005), batch size 64, and experimental settings across algorithms. Hyperparameters are tuned for each method and dataset. We vary the Dirichlet distribution Lin et al. (2020) coefficient $\beta$ to control data heterogeneity (non-IID to IID). For vision tasks, we employ a CNN(batch normalization) model, consistent with the baselines. A fixed random seed (42) is employed to ensure consistent client selection and mini-batch generation across experiments, thus providing a controlled environment for fair comparisons. Moreover, this setting guarantees the reproducibility of the experiments by mitigating the effects of stochastic variations during both client sampling and local updates.

**Algorithm 1:** The Algorithm of KpFL

**Input:** Dataset $\{D_1, D_2, \ldots, D_N\}$, learning rate $\eta$, communication rounds $R$.

**Output:** Personalized models $\{\hat{\theta}_1, \hat{\theta}_2, \ldots, \hat{\theta}_N\}$.

**Initialize:** $\theta_i, \phi_i, v_i$;

// Server-side:

**for** $r \leftarrow 1$ **to** $R$ **do**

   **for** *client $i$ **in parallel*** **do**

      $\hat{\theta}_i^{(t+1)} \leftarrow (\theta_i + \Delta_i) \cdot h_{\text{KAN}_i}(v_i^{(t)}, \phi_i^{(t)})$;

      // Get critical block $K$

      Set $\hat{\theta}_i^{(t+1)}$ in $\hat{\theta}_i^K$;

      $\Delta \theta_i \leftarrow \text{CLIENTUPDATE}(\hat{\theta}_i^{(t+1)})$;

      Update $\theta_i^{t+1} \leftarrow \{\theta_1, \theta_k, \ldots, \theta_n\}$ with $\Delta \theta_i$;

      Update $v_i^{t+1}$ and $\phi_i^{t+1}$ with Equation 13, 12;

// Client update:

Client $i$ receives $\hat{\theta}_i^{(t+1)}$ from the server;

**for** *each local epoch do* **do**

   **for** *mini-batch $\xi_t \subseteq D_i$* **do**

      // Local Training:

      $\theta_i \leftarrow \theta_i - \eta \nabla_{\theta_i} L_i(\theta_i; \xi_t)$;

**return** $\Delta \theta_i$;

