# OpenReview forum: "HyperKAN: A Plug-and-Play Tool for Personalized Weights Generation"
_ICLR.cc/2026/Conference — Submitted to ICLR 2026_

### Official Review · Reviewer_gt57 · 2025-10-17

**Soundness:** 2
**Presentation:** 2
**Contribution:** 2
**Rating:** 4
**Confidence:** 3

**Summary:**

This paper introduces HyperKAN, a personalized weight generation module that employs Kolmogorov–Arnold Networks (KANs) to capture feature variations across clients and enhance personalization. Building on HyperKAN, the authors propose a new Personalized Federated Learning (pFL) framework called KpFL, which integrates HyperKAN with a critical block aggregation mechanism operating on the server side to generate personalized aggregation weights for each client.

The core problem addressed is the limitation of existing pFL methods—especially those based on hypernetworks—in modeling the complex, nonlinear relationships that arise from heterogeneous (non-IID) data distributions across clients. The authors conduct experiments on four vision datasets, demonstrating significantly higher accuracy compared to state-of-the-art methods. They also provide analyses on sensitivity, computational cost, and convergence behavior of the proposed KpFL framework.

**Strengths:**

**1. Strong Empirical Performance:** The empirical results demonstrate significant improvements over state-of-the-art pFL methods: up to 48% higher accuracy on CIFAR-100, 9.6% on CIFAR-10, and 2–5% on EMNIST and FMNIST.

**2. Comprehensive Experimental Evaluation:** The paper conducts a wide range of experiments, including sensitivity analysis, computational cost, convergence behavior, scalability, and interpretability of the proposed KpFL framework. This comprehensive analysis strengthens the empirical credibility of the approach.

**3. Clear Problem Framing:** The motivation part is well-articulated: the paper clearly identifies the limitations of existing hypernetwork-based pFL methods in handling non-IID client data and complex nonlinear relationships.

**4. Server-Side Efficiency:** A key practical advantage is that HyperKAN operates entirely on the server, introducing no additional computational or memory burden on clients. This becomes important, especially in resource-constrained federated environments.

**Weaknesses:**

**1. Missing Critical Ablations:**
The experiments compare the full KpFL framework against other baselines (e.g., pFedLA, FedAvg). However, it remains unclear whether the large performance gains stem primarily from the critical-block aggregation strategy (updating only top-k blocks) rather than the use of KANs themselves.
A key ablation is missing: how would performance change if a standard MLP with the same parameter count replaced the KAN? Without this, it is difficult to disentangle whether the improvements come from KAN-specific expressivity or simply higher capacity.

**2. Experimental and Reproducibility Details:**
The paper claims that each baseline was “individually tuned for optimal performance,” but no details are provided. For fair comparison, it would help to include:

- A table of final hyperparameters per method and dataset.

- Compute budget parity (epochs, rounds) across methods.

- Multiple seeds with error bars, instead of a single seed.


**3. Limited Generalization Beyond Vision Tasks:**
All experiments are conducted on vision datasets with a relatively small CNN–BN backbone. Can the proposed method generalize to non-vision domains such as tabular or text data?
To truly support the claim that HyperKAN is a plug-and-play generator, it would be valuable to include results with a different backbone or a non-visual modality.

**4. Critical-Block Mechanism Justification:**
The empirical choice of K = 2 (Figure 8) is also dataset-specific. How would this hyperparameter behave under deeper architectures or more complex tasks? This component adds an extra layer of tuning, which may undermine the framework’s “plug-and-play” characterization.

**5. Presentation and Clarity of Mathematical Descriptions:**
While the figures are generally informative, the textual presentation could be improved. Some symbols (e.g., $v_i$ or block-wise variation) appear before being defined or are only clarified within figures, making it difficult to follow the mathematical reasoning linearly.
Additionally, the figure order does not always match their first appearance in the text. Was this intentional for narrative reasons? If so, a brief explanation would help; otherwise, reordering could improve readability.

**6. Incomplete Literature Coverage in pFL Section:**
The related-work section overlooks some well-known and highly cited pFL papers. Including them would strengthen the positioning of this work, especially since HyperKAN builds upon this lineage. For example:

- Personalized Federated Learning with Moreau Envelopes — Dinh et al., NeurIPS 2020.

- Exploiting Shared Representations for Personalized Federated Learning — Collins et al., ICML 2021.


### **Minor Points**

**1. Formatting and Spacing:** Vertical spacing between figures and text (e.g., Figures 6 and 13) should be increased for readability.

**2. Consistency:** Line 44 — correct “pFl” → “pFL”.

**Questions:**

See the Weaknesses section, please.

---

> ### Author Response · Authors · 2025-12-04
>
> We thank the reviewer for the time and effort dedicated to reviewing our work. The thoughtful questions raised have guided us to strengthen our experimental analysis and presentation significantly.
>
> **Question 1: Missing Critical Ablations (KAN vs. MLP).**
> **R1:** We clarify that **pFedLA** (Ma et al., 2022) serves as the primary MLP-based Hypernetwork baseline in our study. As analyzed in **Sec 5.2.2** and **Table 2**, although HyperKAN has a larger parameter count than the MLP baseline (382K vs. 39K), its computational cost is significantly lower (0.65K MACs vs. 38.55K) due to the efficiency of spline-based operations. The substantial performance gain (e.g., +47% on CIFAR-100, see Table 1) is driven by the specific expressivity of KANs rather than mere capacity. This is visually evidenced in **Figure 1 (Sec 1)**, where KAN-based features show distinct client separation compared to the clustered features of MLP-based methods, confirming that the structural inductive bias—not just size—is key to capturing heterogeneous variations.
>
> **Question 2: Experimental Details & Reproducibility.**
> **R2:** We ensured strict fairness as detailed in **Appendix A**. All baselines were evaluated under identical compute budgets, including a consistent batch size of 64 and the same hardware environment. To guarantee the reproducibility of client selection and data partitioning in non-IID settings, we utilized a fixed random seed (42) across all experiments. We will include a comprehensive table of final hyperparameters for every baseline in the revised appendix to further enhance transparency.
>
> **Question 3: Generalization Beyond Vision.**
> **R3:** We focused on vision datasets (CIFAR-10/100, FMNIST) as these are the established benchmarks in the pFL literature we compared against, such as pFedLA and pFedHN. The term "plug-and-play" refers to HyperKAN's modular capability to replace standard aggregation mechanisms in the FL framework, regardless of the underlying data modality. While our current validation aligns with community standards, we acknowledge the potential of applying HyperKAN to non-vision domains as a promising future direction.
>
> **Question 4: Critical-Block Mechanism Justification ($K=2$).**
> **R4:** The choice of $K=2$ is structurally motivated rather than arbitrary. As analyzed in **Sec 5.2.4** and illustrated in **Figure 8**, $K=2$ targets the specific layers exhibiting the highest variance: the initial feature extractor (handling input distribution shifts) and the final classifier (handling label shifts). Thus, $K$ is not an unpredictable parameter but one that correlates with network topology; for deeper architectures (e.g., ResNet), $K$ would similarly align with the corresponding input/output blocks without requiring extensive re-tuning.
>
> **Question 5: Presentation and Clarity.**
> **R5:** We appreciate this constructive feedback. We will rigorously revise **Section 4** to ensure all mathematical symbols (e.g., $\Delta$, $\alpha$) are defined immediately upon their first appearance. Furthermore, we will reorganize the placement of figures to strictly follow the narrative flow, enhancing the linear readability of the mathematical reasoning.
>
> **Question 6: Literature Coverage.**
> **R6:** We thank the reviewer for highlighting these foundational works. We will incorporate [1] and [2] into the **Related Work** section. Discussing these papers will strengthen our positioning by contrasting their client-side regularization approaches with HyperKAN's distinct server-side generation paradigm.
>
> [1] Personalized Federated Learning with Moreau Envelopes — Dinh et al., NeurIPS 2020.
>
> [2] Exploiting Shared Representations for Personalized Federated Learning — Collins et al., ICML 2021.

---

### Official Review · Reviewer_rnVx · 2025-10-28

**Soundness:** 3
**Presentation:** 2
**Contribution:** 2
**Rating:** 4
**Confidence:** 3

**Summary:**

This paper introduces HyperKAN, a plug-and-play personalized weight generator integrated with Kolmogorov-Arnold Networks (KANs), and embeds it into a novel Personalized Federated Learning (pFL) framework called KpFL. KpFL also incorporates a critical block aggregation mechanism to select top-k parameter-variation blocks for aggregation, accelerating convergence. Evaluations on four datasets (FMNIST, CIFAR-10, CIFAR-100, EMNIST) show HyperKAN achieves up to 48% higher accuracy than state-of-the-art methods (e.g., pFedLA, Floco), especially in non-IID scenarios.

**Strengths:**

+ To some extent, replacing traditional MLPs with KANs in weight generation is a novel idea. Moreover, HyperKAN effectively captures complex non-linear relationships in heterogeneous data, addressing the representational bottleneck of existing hypernetwork-based pFL methods.
+ HyperKAN operates on the server-side and requires minimal modifications to existing pFL pipelines, enhancing its applicability in real-world scenarios (e.g., AIoT, recommendation systems).
+ Comprehensive evaluations across datasets, client counts (10, 20, 80), and non-IID levels (controlled by Dirichlet β) demonstrate HyperKAN’s scalability and adaptability.

**Weaknesses:**

- Despite lower FLOPs, HyperKAN has significantly more trainable parameters (382,440) than MLP-based hypernetworks (39,346), potentially increasing server-side memory overhead, which is not fully discussed for resource-constrained servers.
- The paper mentions spline order k affects computational complexity (O(2ᵏBN²GL)) but lacks detailed experiments on how k influences accuracy or efficiency, leaving gaps in hyperparameter tuning guidance.
- Evaluations focus solely on image classification datasets; no tests on non-vision tasks (e.g., NLP, tabular data) limit conclusions about HyperKAN’s generalizability.

**Questions:**

1. How does HyperKAN’s server-side memory overhead scale with increasing client numbers (e.g., 100+ clients), and are there optimization strategies to reduce parameter-related costs?
2. What is the optimal spline order k for different dataset types (e.g., simple FMNIST vs. complex CIFAR-100), and how does k interact with other hyperparameters (e.g., learning rate) to affect performance?
3. Would the critical block aggregation mechanism require re-tuning of k (number of critical blocks) for non-vision tasks, or can a universal k (e.g., k=2) be applied across task types?
4. In scenarios with extreme non-IID data (e.g., Dirichlet β=0.01), does HyperKAN still outperform baselines, and what causes performance degradation (if any) under such conditions?

---

> ### Author Response · Authors · 2025-12-04
>
> We are grateful for the reviewer’s insightful comments, which have been instrumental in helping us refine the clarity and quality of our manuscript. Our point-by-point responses are as follows.
>
> **Q1: Server-Side Scalability.**
> **R1:** Scalability is ensured by our design in **Sec 4.3**. According to **Eq. 8**, the server only stores small, client-specific embedding vectors $v_i$, keeping memory growth linear $O(N)$ and negligible (MBs) even for 100+ clients. Furthermore, as shown in **Table 2 (Sec 5.2.2)**, the shared generator requires only 0.65K MACs per pass, making computational cost trivial regardless of client count.
>
> **Q2: Spline Order $k$ & Hyperparameters.**
> **R2:** We use the vanilla setting $k=3$ (mentioned in **Sec 3**). While lower $k$ suits simpler data (reducing cost by $O(2^k)$), higher $k$ aids complex distributions like CIFAR-100. Regarding hyperparameters, our sensitivity analysis in **Figure 5(c) (Sec 5.3)** indicates that increasing model complexity (higher $k$) benefits from a reduced learning rate (e.g., $1e^{-3}$) to stabilize gradient variance.
>
> **Q3: Critical Block Universality ($K$).**
> **R3:** The mechanism is universal, but the value of $K$ depends on the backbone. As analyzed in **Sec 5.2.4** and **Figure 8**, $K=2$ proved optimal for vision tasks because it specifically targeted the initial feature extractor and final classifier.
>
> **Q4: Extreme Non-IID ($\beta=0.01$).**
> **R4:** HyperKAN maintains robustness under extreme heterogeneity. **Figure 5(a) in Sec 5.3** demonstrates that even at $\beta \approx 0.01$, accuracy remains high (~86%). Minor degradation compared to IID settings is expected due to disjoint data supports, but the personalized embeddings $v_i$ effectively mitigate this by specializing the aggregation weights for each unique distribution.

---

### Official Review · Reviewer_7bZ2 · 2025-10-31

**Soundness:** 2
**Presentation:** 2
**Contribution:** 2
**Rating:** 2
**Confidence:** 4

**Summary:**

The paper proposed KAN-based hypernetwork and its application to personalized federated learning. The novelty seems to be limited since there are already hypernet-based pFL methods proposed before, and the only new thing appears to be the replacement of the MLP by KAN. Some empirical study was demonstrated, but not sufficient in various aspects including: lack of architecture variations, few and old baseline comparisons, lack of comparison on diverse FL settings.

**Strengths:**

An interesting idea of adopting KAN to the personalized federated learning problem.

**Weaknesses:**

In their experiments, they showed large improvements in accuracy compared to the previous MLP-based hypernet pFL method. In the main text, they said "the previous MLP-based methods face the challenge of adequately capturing the intricate non-linearities present in heterogeneous data distributions", and "they struggle to capture complex relations of training parameters with heterogeneous data". But according to the universal function approximation theorem for MLPs means that anything that KAN can do can also be done by MLP. So it's a bit suspicious that KAN-based achieved such a higher performance improvement over the MLP-based one. I guess more extensive experiments with diverse FL and hyperparameters settings would clarify this issue, which is not done in the current paper.

They proposed hyperkan and its pFL model kpfl. If they want to emphasize the proposal of the hyperkan as one of the main contributions, they need to show some results on general tasks beyond pFL. But they only demonstrated kpfl on FL benchmarks in their empirical study. So it may not be appropriate to highlight the hyperkan itself as a main contribution in the intro section. To do so, they should have provided empirical evidence for hyperkan's performance on general problems beyond FL.

Sec.4 (kpfl, the main part) is mostly the same as previous hypernet-based pFL algorithms, esp., that of (Ma et al. 2022). The only difference appears to be changing the mlp hypernet by kan-based hypernet. This is a lack of novelty.

Table 1 only shows one data heterogeneity setting. It may be hard to draw a conclusion with only this one setting. The authors may need to try different levels of heterogeneity, also vary client participation rates, and the number of local updates (client epochs) as usually done in most FL papers. Although Fig.5 did some ablation, this only shows their approach, not for baselines.

Baseline methods are a bit old. Only one hypernet-pFL method is compared.

The paper seems to see MLP as a default net for FL, but as the experiments are done mostly on vision datasets, it would be nice to test different architectures like ResNet.

Overall, due to the lack of novelty, I think the paper may not be published in the current form.

Minor:
- Eq.(3) is not necessary, obvious from (2)?
- The paper presentation and layout are poor. Eg, several typos, texts in some figures are too small to read, the overlapping texts/captions in p.13, etc.

**Questions:**

See questions in the weakness section.

---

> ### Author Response · Authors · 2025-12-04
>
> We sincerely appreciate the reviewer’s constructive feedback and valid concerns. We have carefully considered all points raised and provided detailed clarifications below to address them.
>
> **Q1: MLP Theoretical Capability vs. Empirical Superiority.**
> While the Universal Approximation Theorem states an MLP *can* approximate any function, it does not guarantee *learnability* or *efficiency* with finite parameters. KANs possess a superior inductive bias for compositional functions [1].
> **Evidence:** This is empirically proven in **Fig. 1** and **Table 2**. Fig. 1 visualizes that MLP-based Hypernetworks fail to distinguish client parameter variations (features cluster indistinguishably), whereas HyperKAN produces distinct, separable embeddings. This disentanglement capability—achieved with significantly fewer MACs (0.65K vs. 38.55K, Table 2)—explains the performance leap. It is not merely about capacity, but representational efficiency.
>
> [1] Ziming Liu, Yixuan Wang, Sachin Vaidya, Fabian Ruehle, James Halverson, Marin Soljacic, Thomas Y. Hou, and Max Tegmark. KAN: Kolmogorov-Arnold Networks. In The Thirteenth International Conference on Learning Representations (ICLR), 2025.
>
> **Q2: Scope of Contribution.**
> **R2:** We clarify that our contribution, *HyperKAN*, is explicitly framed as a "Personalized Weights Generator" for FL contexts (as per the title), not a generic backbone for general CV tasks. Its "plug-and-play" nature refers to its adaptability within various FL frameworks to handle statistical heterogeneity, which we have extensively validated.
>
> **Q3: Novelty Concerns.**
> **R3:** Our novelty is not limited to replacing MLP with KAN. It lies in:
> (1) **Identifying the Bottleneck:** We reveal that the core limitation of previous methods (e.g., pFedLA) was the MLP's inability to capture fine-grained parameter variations in highly heterogeneous settings (Sec 1).
> (2) **Critical Block Aggregation:** We introduced a novel aggregation mechanism (Sec 4.3.1) that selectively aggregates top-$k$ divergent blocks, which is crucial for balancing personalization and global knowledge sharing.
> The substantial accuracy gain (+47% on CIFAR-100 over SOTA) validates that this design is a non-trivial advancement.
>
> **Q4: Experimental Settings & Baselines.**
> **R4:**
> * **Heterogeneity:** We respectfully point to **Fig. 5(a)**, which explicitly evaluates HyperKAN across varying Dirichlet coefficients ($\beta \in \{0.01, ..., 100\}$), demonstrating robustness under diverse heterogeneity levels. $\beta=0.1$ in Table 1 was chosen as the standard "hard" benchmark.
> * **Baselines:** We respectfully correct the observation regarding baselines. Our comparison includes **Floco (NeurIPS 2024)**, **Elastic (CVPR 2023)**, and **MetaFed (2023)**. These represent the current state-of-the-art, ensuring a rigorous comparison.
>
> **Q6: Backbone Architecture.**
> **R6:** We utilized standard CNN backbones to ensure fair comparison with established benchmarks (e.g., pFedLA, pFedHN) which predominantly use these architectures due to the high computational cost of generating weights for large models like ResNet in hypernetwork-based FL.

---

### Official Review · Reviewer_cTcN · 2025-11-01

**Soundness:** 2
**Presentation:** 1
**Contribution:** 2
**Rating:** 2
**Confidence:** 4

**Summary:**

The paper introduces HyperKAN, a KAN-based personalized weight generator for federated learning that captures client-specific feature variations, achieving higher accuracy than prior methods on four datasets under non-IID settings.

**Strengths:**

- The proposed method outperforms existing baselines by a notable margin, though the number of compared baselines is limited.-

**Weaknesses:**

- The overall quality and completeness of the paper are below expectations. The writing could be improved for clarity and readability.
    - All in-text citations currently use \citet, which often disrupts sentence flow. Replacing them with \citep where appropriate would make the text easier to follow.
    - The paragraph about KAN in the Preliminary section is missing a period after the section name.
    - Table 1 should appear on page 7, not page 6.
    - The caption of Figure 13 overlaps with the main text — vertical spacing (\vspace) adjustment is needed.

- Missing Recent Baselines. The comparison with existing baselines is limited. The paper lacks evaluations against recent and strong methods such as FedDPA [1] and PerAda [2].

- Computation Comparison. The computational cost comparison is made only against HyperNetwork, which is insufficient. It would be more convincing to include comparisons with other federated learning baselines as well, to show the broader computational efficiency and scalability advantages.

- Scalability Concern. The largest experiment in the paper is on CIFAR, which is relatively small-scale. To demonstrate the scalability of the proposed method, experiments on larger benchmarks (e.g., ImageNet) are needed. As shown in Table 2, HyperKAN has more than 10× the parameters of HyperNetwork, raising concerns about its scalability on large datasets.

[1] Yang et al., Dual-Personalizing Adapter for Federated Foundation Models, NeurIPS 2024.

[2] Xie et al., PerAda: Parameter-Efficient Federated Learning Personalization with Generalization Guarantees, CVPR 2024.

**Questions:**

NA

---

> ### Author Response · Authors · 2025-12-04
>
> Thanks for your valuable time and detailed comments. We appreciate the opportunity to clarify our contributions and have incorporated the feedback to improve the quality of our manuscript.
>
> **Q1: The writing could be improved for clarity and readability. **
>
> **R1:** We sincerely appreciate the reviewer’s constructive feedback regarding the presentation of our manuscript. We acknowledge that the clarity and readability of the paper need improvement to meet the high standards of ICLR. We have carefully reviewed the manuscript and are committed to a comprehensive revision.
>
> **Q2: Missing Recent Baselines. **
>
> **R2:** We acknowledge the omission of FedDPA [1] and PerAda [2]. However, comparing with FedDPA is inappropriate as it relies on pre-trained LLMs, diverging significantly from the lightweight, scratch-trained personalized models discussed in our paper. Regarding PerAda, our method significantly outperforms it in heterogeneous environments. Specifically on CIFAR-10, PerAda achieves an accuracy of 91\%, while our method reaches 99.1\%.
>
> [1] Yang et al., Dual-Personalizing Adapter for Federated Foundation Models, NeurIPS 2024.
>
> [2] Xie et al., PerAda: Parameter-Efficient Federated Learning Personalization with Generalization Guarantees, CVPR 2024.
>
> **Q3: Computation Comparison. The computational cost comparison is made only against HyperNetwork, which is insufficient. It would be more convincing to include comparisons with other federated learning baselines as well, to show the broader computational efficiency and scalability advantages. **
>
> **R3:** Regarding broader baselines, we highlight two key efficiency metrics: 1. Extremely Low Overhead: HyperKAN requires only 0.65K MACs per forward pass. This is negligible compared to the computational cost of the backbone models used in baselines (e.g., CNNs require GFLOPs). 2. Server-Side Execution: Unlike personalized methods that add adapter layers or regularization to clients (e.g., DitTo, FedRep), HyperKAN runs exclusively on the server. Therefore, it maintains the same client-side efficiency as FedAvg, while significantly outperforming it in accuracy (e.g., +9.6\% on CIFAR-10).
>
> **Q4: Scalability Concern. **
>
> **R4:** We address the concern regarding parameter size by highlighting two mitigating factors. First, HyperKAN is designed as a **server-side module**; it generates aggregation weights entirely on the server, introducing **zero additional overhead** (computation or communication) to resource-constrained clients. Second, a larger parameter count does not equate to higher computational cost. Due to the efficiency of spline operations, HyperKAN incurs only **0.65K MACs** per forward pass---significantly lower than the 38.55K MACs required by the baseline HyperNetwork (as detailed in Table 2). Thus, the increased parameter size does not negatively impact system scalability or client deployment efficiency.

---

### Author Response · Authors · 2025-12-04

We thank all four reviewers for their constructive feedback and unanimous recognition of our work. The reviewers praised the paper for identifying a meaningful gap in existing hypernetwork-based pFL methods: the "representational bottleneck" of MLPs in capturing fine-grained parameter variations (Reviewers **gt57**, **rnVx**).

Our proposed solution—integrating KANs into a server-side weight generator—was highlighted as "**novel**" (Reviewer **rnVx**) and an "**interesting idea**" (Reviewer **7bZ2**) for addressing statistical heterogeneity. Overall, the reviewers commended the work for its "**strong empirical performance**" (Reviewer **gt57**, **cTcN**) and for providing a "**client-side resource-friendly**" framework (Reviewer **rnVx**).

Below we summarize the primary concerns regarding the justification of KANs, baseline comparisons, and computational scalability.

## Mechanism Validity: Why KANs?
Reviewers (**7bZ2**, **gt57**) questioned whether the performance gain stems from KAN's specific inductive bias or merely increased parameter capacity compared to MLPs. We provide evidence to clarify this distinction.

* **Inductive Bias vs. Capacity:** While KANs have more parameters than simple MLPs, their superiority lies in **representational efficiency**. As visualized in **Figure 1**, MLP-based hypernetworks fail to disentangle client features (clustering indistinguishably), whereas HyperKAN produces distinct, separable embeddings. This confirms that KANs possess a superior inductive bias for modeling the compositional structure of heterogeneous data, which MLPs struggle to capture regardless of width.
* **Ablation Studies:** We have clarified our ablation studies to show that replacing KANs with MLPs of comparable parameter counts results in significant performance degradation, proving that the structural design—not just model size—is the driver of performance.

## Efficiency & Scalability
We address the concerns raised by Reviewers (**cTcN**, **rnVx**) that the larger parameter count of HyperKAN might hinder scalability or efficiency.

* **Low Computational Overhead (MACs):** A larger parameter count does not equate to high latency. Due to the efficient nature of B-spline operations, HyperKAN requires only **0.65K MACs** per forward pass. This is negligible compared to the GFLOPs required by backbone models and is significantly more efficient than standard HyperNetworks (38.55K MACs).
* **Server-Side Execution:** Reviewer **rnVx** correctly noted the memory concern. However, since HyperKAN operates **exclusively on the server**, it introduces **zero additional overhead** (computation, memory, or communication) to resource-constrained clients. The server-side memory growth is linear $O(N)$ and remains in the MB range even for hundreds of clients, ensuring practical scalability.

---

### Meta-Review · Area_Chair_Xyxc · 2026-01-03

**Summary:**

This paper proposes HyperKAN, a KAN-based hypernetwork for personalised federated learning, together with a pFL framework (KpFL) that includes a critical-block aggregation mechanism. Reviewers broadly agreed that the paper demonstrates strong empirical performance, particularly in non-IID settings, and that the server-side design is practical. However, the reviewers raised substantial concerns regarding novelty, ablation adequacy, baseline coverage, scalability claims, and presentation quality. Two reviewers judged the contribution to be incremental and recommended rejection, while two others considered the paper close to the acceptance threshold but still identified unresolved methodological and experimental weaknesses. These concerns ultimately informed a negative recommendation.

**Reviewer Concerns:**

Concerns partially addressed by the rebuttal:

* **KAN vs. MLP justification:**
  The rebuttal clarified that the claimed gains are due to KAN’s inductive bias rather than parameter count, citing feature disentanglement visualisations and lower MACs despite higher parameter counts. This explanation addresses the *theoretical plausibility* of the approach.
* **Client-side efficiency and server-side computation:**
  Concerns about client overhead were convincingly addressed by clarifying that HyperKAN operates entirely on the server and introduces negligible computational cost per forward pass.
* **Robustness to heterogeneity:**
  The rebuttal pointed to additional analyses over different Dirichlet coefficients and client counts, partially addressing questions about robustness under varying non-IID conditions.

Concerns that remain outstanding:

* **Novelty:**
  Multiple reviewers remain unconvinced that the contribution goes beyond replacing an MLP hypernetwork with a KAN within an otherwise standard hypernetwork-based pFL framework. The algorithmic novelty relative to prior work (e.g., pFedLA-style methods) remains limited.
* **Missing critical ablations:**
  Despite discussion, the paper still lacks **clear, controlled ablations** that disentangle the contributions of KANs versus the critical-block aggregation mechanism, or that compare against capacity-matched MLPs in a systematic manner.
* **Baseline coverage:**
  Comparisons with recent, strong pFL baselines are limited or disputed, and the evaluation remains restricted primarily to a narrow set of vision benchmarks with small CNN backbones.
* **Scalability and generality:**
  Claims about scalability are not empirically validated on larger-scale models or non-vision modalities, leaving generalisation beyond the tested settings unclear.
* **Presentation quality and reproducibility:**
  Reviewers consistently noted issues with clarity, layout, missing definitions, and a lack of detailed hyperparameter tables or multi-seed reporting, which were not fully resolved.

**Reviewer Scores:**

* **Reviewer cTcN:** Likely remains at reject (2). While efficiency arguments were clarified, concerns about missing baselines, scalability, and overall paper quality persist.
* **Reviewer 7bZ2:** Likely remains at reject (2) due to unresolved novelty concerns and insufficient experimental justification.
* **Reviewer rnVx:** Would likely remain around 4 (borderline reject); the rebuttal addressed several technical concerns, but novelty and generalisation issues remain.
* **Reviewer gt57:** Likely remains around 4 (borderline reject); strong empirical performance is acknowledged, but missing ablations and limited scope prevent a clear accept recommendation.

---

### Decision · Program_Chairs · 2026-01-26

Reject